# Sensitivity and Specificity of a Screening Test for the Detection of Deficiencies in Visuo-Perceptual Skills

**DOI:** 10.3390/brainsci14070705

**Published:** 2024-07-14

**Authors:** Elizabeth Casillas-Casillas, Luis Héctor Salas-Hernández, Katie Lynn Ortiz-Casillas, Tamara Petrosyan, Sergio Ramírez-González, Luis Fernando Barba-Gallardo

**Affiliations:** 1Department of Optometry, Health Science Center, Autonomous University of Aguascalientes, Aguascalientes 20131, Mexico; elizabeth.casillas@edu.uaa.mx (E.C.-C.); hector.salas@edu.uaa.mx (L.H.S.-H.);; 2College of Optometry and East New York Diagnostic and Treatment, State University of New York, Buffalo, NY 14203, USA

**Keywords:** sensibility, specificity, visuo-perceptual skills, vision

## Abstract

This study determines the sensitivity and specificity of a screening test to detect perceptual abnormalities and whether there are differences between gender. Vision is a complex process involving visual perception. Any alterations can affect learning, so having a screening test in Spanish that is easy to use and reliable for timely diagnosis will reduce the percentage of visuo-perceptual interference during learning process. A total of 200 subjects participated, aged between 8 and 15 years old, with good visual acuity, and no strabismus, amblyopia, ocular pathology, or neurological damage. The Petrosyan questionnaire (screening test) was employed to identify symptoms associated with perceptual impairment, and a subsequent assessment was conducted to evaluate perceptual abilities. The mean age was 11.5 years (57% male; 44% female). The screening test indicated that 30% of the subjects were suspected of having perceptual alteration, while 24% were diagnosed with a real alteration in perceptual abilities. The sensitivity was 1 and the specificity was 0.92. The Spanish version of the Petrosyan questionnaire has high sensitivity and specificity values and is therefore considered very accurate for identifying the need for a perceptual assessment. There are statistically significant differences in perceptual abilities according to gender. The female group shows more symptomatology and a higher percentage of alteration in perceptual skills.

## 1. Introduction

Perception is an active process involving the localization and extraction of information from the external environment. Perceptual systems organize this information, engaging in data search and retrieval to develop awareness of surroundings and self. Through perception, individuals attribute meaning to sensory input received from external senses (vision, smell, hearing, touch, and taste) and internal senses (vestibular and kinesthetic systems). The brain interprets and classifies this information, leading to the formation of both simple and complex cognitive concepts. Visual perception, notably, significantly influences learning and academic performance by aiding in the encoding, decoding, and spatial-temporal structuring of letters and information. This article explores the intricate processes underlying perception and its impact on cognitive functioning [1,2].

The visuo-perceptual skills are the prerequisite for more complex conceptual notions that involve reading and writing. They facilitate cognitive development and are divided into three systems: visuo-spatial, which enables a better understanding of directional concepts (up, down, right, left) that organize the external visual space, subdivided into bilateral integration, laterality and directionality; visual analysis, which facilitates recognizing, remembering, and manipulating visual information; and the visual–motor system, which coordinates visual processing skills with motor skills. It is the integration of shape perception with the fine motor system to reproduce more complex visual patterns [3,4].

Deficits in perceptual abilities can result in confusion regarding the identification of letters, difficulty in reading, memory, and recognition of words and numbers, as well as an increased susceptibility to distraction. This can lead to a lack to interest in reading, writing, and analytical tasks due to an inability to effectively process visual information [3,4].

In Spain, it has been reported that 15 to 20% of children with learning disabilities have perceptual problems that interfere with school development and performance [5].

In Colombia, a study determined a prevalence of 30% of alterations in the visuo-perceptual skills of visuomotor integration, saccadic movements, visual attention, and reading and writing process in children aged 6 to 7 years, so that reading and writing allows the individual to acquire learning, making use of processes such as perception, memory, consciousness, and cognitive processes [6,7,8].

There are several factors that have a negative impact on school performance, including visual health problems and visuo-perceptual skill impairments [9].

According to the Organization for Economic Cooperation and Development (OECD), of the 4.5 million 15-year-old students, 1 in 4 reach the basic level of performance in reading, mathematics, and science. In Mexico, according to the PISA (Program for International Student Assessment) test, which evaluates the areas of mathematics, reading, and science, 55% (941,644) of 15-year-old students show a low performance level in at least one of the areas [10].

Because of the above and given that perceptual skills are involved in the learning process, it is necessary for teachers to know the state of perceptual development of students, especially those with school performance difficulties.

Although there are techniques for the evaluation and diagnosis of a perceptual alteration that are specific and exhaustive, they require time, materials, and economic resources, so having a quick tool of minimum cost and easy application that allows screening would be very useful. Therefore, the Petrosyan Visual Perceptual Questionnaire is a tool with all these qualities, consisting of 22 statements, on a Likert scale that was designed by Dr. Tamara Petrosyan, a professor at the College of Optometry at New York University (SUNY), with the purpose of identifying in a timely manner the symptomatology related to any alteration in reading, writing, and/or analysis of information. The maximum score is 88 points; if more than 27 points are obtained, it suggests impairment in information analysis, reading, and/or writing skills. The translation process of the questionnaire Petrosyan followed the procedure established by Brislin [11]. Initially, the text was translated from the original language (American English) into Spanish by experts. Subsequently, ten Spanish-speaking optometrists skilled in perceptual and binocular vision reviewed it. They were provided with the original questionnaire in Spanish and asked to translate it back into English. The translations from all 10 optometrists were compared to the original questionnaire, and coincidences and non-coincidences were detected. To determine the level of similarity, the criteria for interpretation and grammar were considered. Each aspect evaluated was assigned a score from 0 to 3, where 0 = none (if the interpretation was completely different), 1 = low (if there were more different words than similar or identical words), 2 = partial match (if there were more similar/identical words than different words), and 3 = identical (if the words were the same). This scoring system facilitated a quantitative assessment of the fidelity of the translated versions compared to the original questionnaire.

The level of coincidence was determined to be 96.04% with a Cronbach’s Alpha of >95% with respect to the original English version (Figure 1).

For a clinical diagnostic test or screening test to be applied either by support staff, teachers in the classroom, or an eye care professional with a high degree of accuracy in the detection of visual perceptual abnormalities, it needs to be analyzed to determine the sensitivity, specificity, and positive and negative predictive values to classify healthy and unhealthy subjects, as well as the reproducibility of the results [12,13,14]. The objective of this study was to determine the degree of sensitivity and specificity of the Petrosyan questionnaire for the detection of deficiencies in perceptual skills in 200 subjects, ages 8 to 15 years old, and additionally to establish if there are differences in the perceptual skills by gender.

## 2. Methods

An observational, analytical study was carried out in a population of 200 students aged 8 to 15 years with good visual acuity, which was determined for each eye with the Snellen chart in far and near vision. This study was reviewed and approved by the Human Research Ethics Committees in our university. All research was conducted in accordance with the Declaration of Helsinki with all participants providing informed consent signed by the parents of all the children. Patients with strabismus, amblyopia, ocular pathology were excluded. The Petrosyan questionnaire was applied in Spanish by support personnel, and without knowing the results of the questionnaire, qualified optometrists evaluated the perceptual skills with the gold standard test, Motor-Free Visual Perceptual Test Fourth Edition (MVPT-4) (e.g., Author: Ronald P. Colarusso/Donald D. Hammill) [15]. In multiple-choice format, it assesses visual perceptual ability in five skills: visual discrimination, spatial relationship, figure background, and visual closure. It is applied to age groups from 4 to 80 years. It consists of 45 images; it is a valid and reliable test. The standardization of the MVPT-4 test was administered by 88 examiners to 2160 individuals between the ages of 4 years and geriatric adult at 85 sites across the United States. To assess reliability of test items, Cronbach’s coefficient alpha was computed for the overall sample and for each group in the standardization sample this value was 0.80. This value indicates that the MVPT-4 overall is internally consistent. To assesses validity, 27 individuals were administered the MVPT-4 and TVPS-3, with the results indicating there is a significant correlation between both tests. The results suggest that the MVPT-4 adequately measures aspects of visual perceptual skills. Steps in the MVPT-4 test: The examiner presents some images and asks questions about them, the number of correct answers is recorded, the percentile is obtained and compared with what is expected for the age, and a percentile lower than 30 is considered low.

To evaluate the spatial relationship and processing speed, the Primary Mental Abilities test was used (e.g., Author LL. Thurstone T.G. Thurstone), Grade 4–6 level [16]. The test applies to ages 8 to 14. The spatial relationship test requires mental manipulation of a picture. The patient is shown an incomplete picture and is asked to indicate which of 4 possible choices is missing to complete the picture. The correct answer may or may not be in the appropriate spatial orientation. It consists of 25 items and the test must be completed in a maximum time of six minutes. The number of correct answers is noted, the percentile is obtained and compared with what is expected for the age, and a percentile lower than 23 is considered low.

Processing speed was assessed by identifying the similarity between two images, and the test consists of 40 items. The patient is asked to look at a row of four figures and mark the two figures that are the same. The number of correct answers is recorded, the percentile is obtained and compared with the expected for the age, and a percentile lower than 23 is considered low [16].

The writing area was evaluated with the Wold copying sentence test created by Bob Wold to identify visual–motor difficulties in children, it evaluates the ability to integrate fine motor movement guided by vision. It is used to determine speed and accuracy in copying and writing skills. The patient is provided with a sheet of paper with a paragraph written on the top consisting of 110 letters and is asked to write the sentence they observe on the bottom of the sheet as fast as possible; the examiner notes the time it takes to copy the paragraph. The letters copied in one minute are quantified and the result is compared to what is expected for the school grade level [17].

Reading comprehension was assessed using the Readalyzer^TM^ (Compevo AB, Markvardsgatan, Stockholm, Sweden) an eye tracker device consisting of infra-red googles used to determine eye position, which is sensitive to the reflection of light on the cornea. It objectively determines the number of eye fixations and regressions, duration of fixation, speed, and level of reading comprehension. The patient wears glasses and is asked to read a text; at the end of reading the text, 10 questions are asked about the text read, and the patient only has to answer yes or no. Regarding information obtained from the reading comprehension, a percentage of less than 70% of correct answers was considered deficient, and the reading speed was compared with the number of letters per minute expected for the age [18].

The statistical analysis was carried out using descriptive statistics, measures of central tendency and frequency distribution and was classified according to gender. For inferential statistics, normality distribution test was performed using the Kolmogorov–Smirnov and Shapiro–Wilk tests (*p* > 0.05), and Wilcoxon test for nonparametric data (*p* < 0.05). For sensitivity, specificity, and positive and negative predictive values, a binary or 2 × 2 table was constructed from which the different probable results of a screening test were obtained.

## 3. Results

A sample of 200 subjects from two schools in Aguascalientes, Mexico, was chosen. The mean age was 11.54 ± 1.82 years. Overall, 56.5% (113 cases) were male and 43.5% (87 cases) were female. The mean age for the male group was 11.74 ± 1.99 and the mean age for the female group was 11.30 ± 1.82 years.

The results of Petrosyan questionnaire in Spanish revealed that 29.5% of the cases obtained a score higher than 27 points; this score suggests an evaluation of perceptual skills. The male group 23.90% presented symptoms with a median of 18, and 36.78% of the female group with a median of 22. The Wilcoxon test for non-parametric samples showed a statistically significant difference (Table 1).

To determine the diagnosis of perceptual skills, the values obtained for each test were analyzed. The results show that 23.5% of the cases presented an alteration in perceptual skills. With respect to gender, 23% of males (26 cases) and 24.13% of females (21 cases) presented an alteration in perceptual skills. The Wilcoxon test shows a statistically significant difference (Table 2).

With respect to the evaluation of perceptual skills (gold test), Table 3 shows the results of the percentages of deficiency in performance for each test. It can be observed that the tests that show a higher percentage of deficiency are reading speed, visual analysis, and reading comprehension.

The Wilcoxon test for nonparametric samples was performed to identify differences in terms of gender. Figure 2 shows that in all variables, there is a statistically significant difference.

To determine sensitivity and specificity, a 2 × 2 binary table was constructed. Table 4 presents the results.

The ROC curve Figure 3 presents the area under the curve with a confidence interval ranging between 0.9205 and 0.9786. The red line allows to identify the area under the curve to identify children who can be identified with visual perception according to their age from those who have an area of the visual perception zones affected (visual analysis, spatial relationship, processing speed, reading comprehension or writing, see Table 5).

## 4. Discussion

The Spanish version of the Petrosyan questionnaire shows that 29.5% of the cases obtained a score higher than 27 (Table 1), which suggests an alteration in perceptual skills, specifically in the areas of reading, writing, and/or information analysis. With respect to gender (Table 2), it is observed that the female group presents more symptoms showing a statistically significant difference (*p* < 0.005).

Regarding the results of the perceptual skills assessment, the results show a lower percentage than the questionnaire, since only 23.5% of the cases (Table 3) were diagnosed with a deficiency in perceptual skills under the criterion of at least three or more diagnostic tests with deficient performance. The skills with the highest percentage of impairment in the total sample were reading speed with 69% of the cases, visual analysis with 35.5%, and reading comprehension with 26%. Comparing the results in relation to gender (Figure 2), females presented a higher percentage of deficiency, thus showing statistically significant differences (*p* < 0.005).

With respect to gender, it is observed that in the male group visual analysis, spatial relationship and writing present a higher percentage of alteration when compared to the female group, while the speed of processing and reading comprehension presented a higher percentage of alteration in the female group, these differences being statistically significant (see Figure 2).

In one study, reading accuracy, speed, and fluency were analyzed in 190 cases, and the results showed that accuracy and fluency were higher in males, while reading speed was higher in females; the results of the study coincide with those of this study, having very similar samples [19].

Another study analyzed reading comprehension in 54 cases, and although they agree with other studies that gender is not associated with the level of reading comprehension, they did find that reading comprehension is slightly higher in females [20]. In this study, reading comprehension is higher in males. These results may be due to the difference in the size of the sample; in addition, the difference found is minimal.

In other research, the prevalence of perceptual alterations was analyzed in 208 cases, and the results showed a prevalence of 30% [6], whereas in this study, a lower prevalence of 23.3% was found, and in this case, the samples are very similar in both studies.

Two previous studies analyzed the importance of calculating sensitivity and specificity, mentioning that there was a high sensitivity when they are greater than 95% [21,22]. In these cases, the sensitivity was 1, and there was a high specificity of between 0.91 and 1 (in this study, it is 0.92).

Diagnostic tests or screening tests are used for different purposes, such as screening a population, ruling out or confirming a diagnosis or monitoring a pathology. For a correct evaluation of a diagnostic test, it is necessary to know the sensitivity and specificity, which are intrinsic characteristics of a test, and they are interdependent since an increase in sensitivity is accompanied by a reduction in specificity and vice versa. In this study, the values obtained show a high value. The ROC curve presents the area under the curve with a confidence interval ranging between 0.9205 and 9786, which confirms its high sensibility and specificity.

## 5. Conclusions

The reliability of the Petrosyan questionnaire in its Spanish version is high since the sensitivity is 1 and the specificity is 0.92. Therefore, the screening test (Petrosyan questionnaire) is a reliable tool to identify the need for a perceptual evaluation in the areas of visual analysis, writing, comprehension, and reading speed.

There are statistically significant differences in perceptual abilities with respect to gender. The female group presents more symptomatology and a higher percentage of impairment in perceptual skills. Visual analysis skills, spatial relationship, and reading and writing speed present a higher percentage of alteration in the male group, while processing speed and reading comprehension present a higher percentage of alteration in the female group. The limitation of this study was that the Primary Mental Abilities test was used outside the age range; however, another test, MVPT-4, which is designed for a wide age range, was used.

## Figures and Tables

**Figure 1 brainsci-14-00705-f001:**
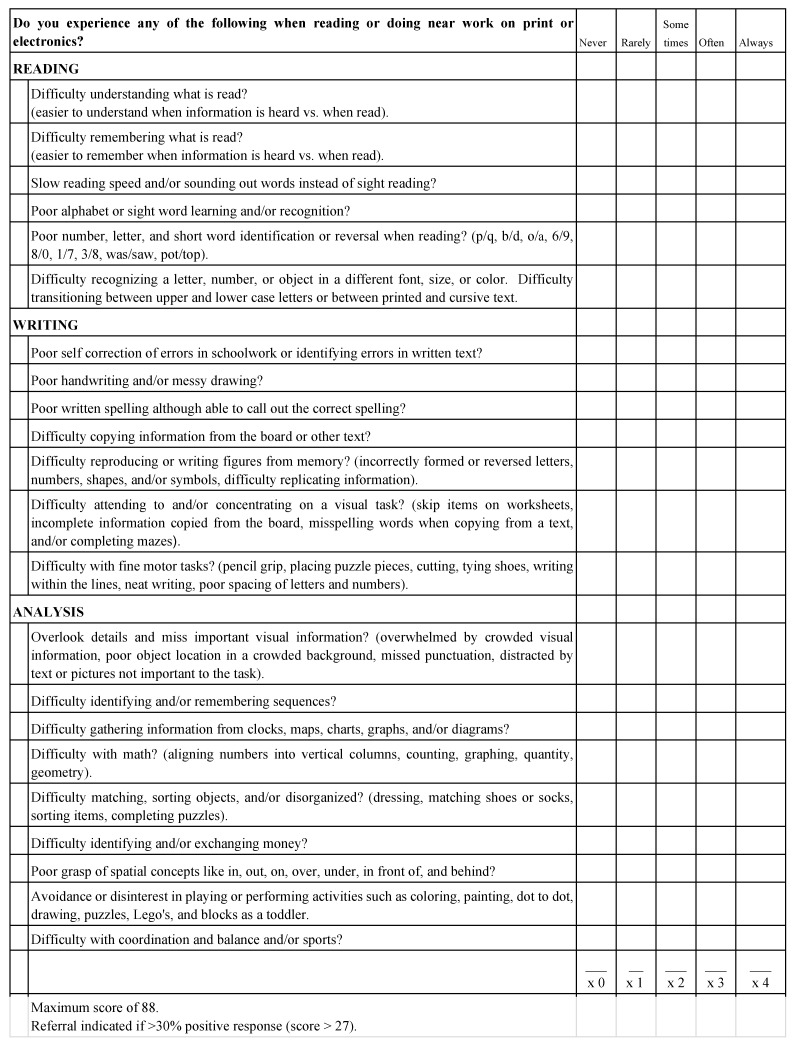
Petrosyan Visual Perceptual Questionnaire.

**Figure 2 brainsci-14-00705-f002:**
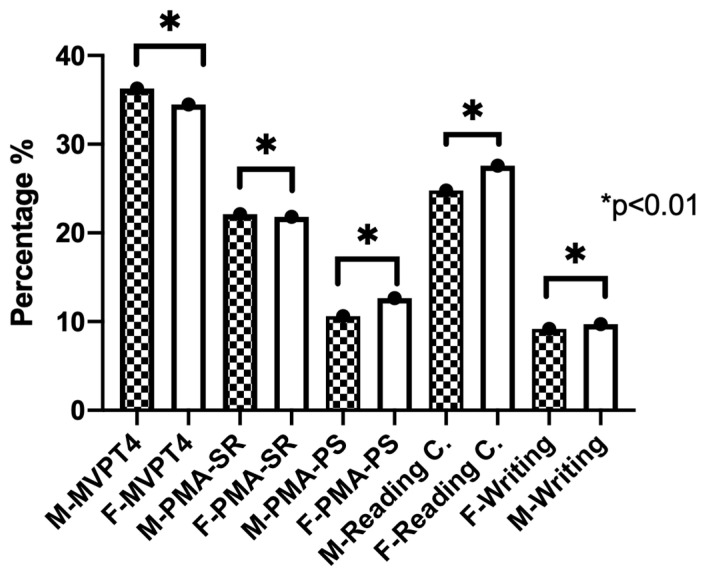
Comparison of perceptual deficiency for each diagnostic test by gender. M—male; F—female; Motor-Free Visual Perceptual Test-4 (MVPT-4); Primary Mental Abilities-spatial relations (PMA-SR); Primary Mental Abilities-perceptual speed (PMA-PS); reading comprehension (Readalyzer) (Reading C.); writing (Wold copy sentence). Wilcoxon signed rank test. *p* < 0.05 significance statistic.

**Figure 3 brainsci-14-00705-f003:**
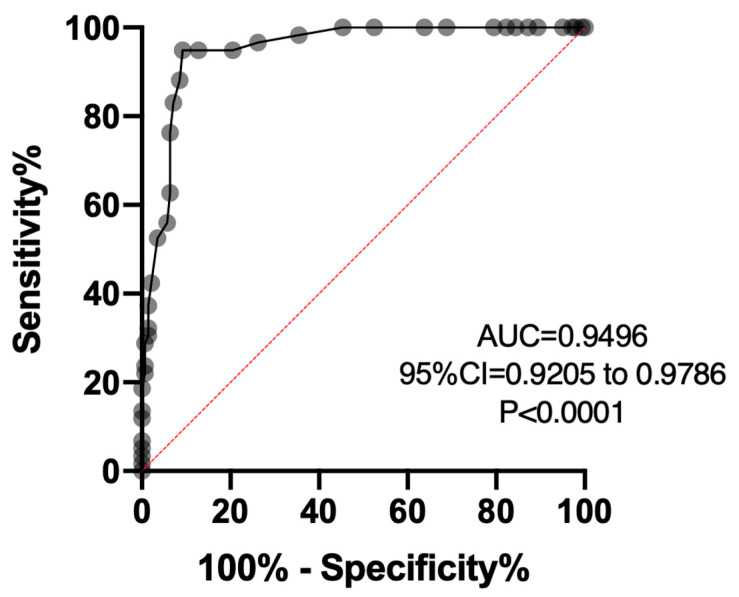
ROC curve—sensibility and specificity; AUC = area under curve; confidence interval, *p* value.

**Table 1 brainsci-14-00705-t001:** Results of Petrosyan questionnaire by gender.

Variable	Median Male Group/Percentage	Median Female Group/Percentage	*p* Value
Petrosyan Questionnaire	18 (23.90%)	22 (36.78%)	<0.001

**Table 2 brainsci-14-00705-t002:** Results alteration in perceptual skills by gender.

Variable	Number of Cases Male Group	Number of Cases Female Group	*p* Value
Diagnosis of perceptual disturbance	26 cases (23%)	21 cases (24.13%)	<0.001

**Table 3 brainsci-14-00705-t003:** Percentage of deficient performance by each diagnostic test: Motor-Free Visual Perceptual Test-4 (MVPT-4), Primary Mental Abilities—spatial relations (PMA-SP). Primary Mental Abilities—perceptual speed (PMA-PS).

Skill Test	Percentage of Deficient Performance
Visual analysis (MVPT-4)	35.5%
Spatial relation (PMA-SR)	22.12%
Processing speed (PMA-PS)	11.50%
Reading comprehension (Readalyzer)	26%
Reading speed (Readalyzer)	69%
Writing (Wold copy sentence)	9.50%

**Table 4 brainsci-14-00705-t004:** Binary table (2 × 2). Sensibility and specificity from Petrosyan questionnaire.

Screening Test	Diagnostic Test		Total
**Positive**	True positive (a) **47**	False positive (b) **12**	59
**Negative**	False negative (c) **0**	True Negative (d) **141**	141
Total	**47**	**153**	**200**
**Indicator**	**Calculation**	**Result**	
**Sensitivity**	a/a + c	47/47 = **1**	
**Specificity**	d/b + d	141/12 + 141 = **0.92**	
**Positive predictive value**	a/a + b	47/47 + 12 = **0.79**	
**Negative predictive value**	d/d + c=	141/141 + 0 = **1**	
**Positive likelihood ratio**	S/1 − E	1/1 − 0.92 = **12.5**	
**Negative likelihood ratio**	1 − S/E	1 − 1/0.92 = **0.086**	
**Accuracy index**	(a + d)/(a + d + b + 0)	(47 + 141)/(47 + 141 + 12 + 0) = **0.94**	

**Table 5 brainsci-14-00705-t005:** Sensibility and specificity by each test.

Indicator	MVPT-4	PMA-SR	PMA-SP	Reading Comprehension	Writing
Sensitivity a/a + c	0.83	1	1	1	1
Specificity d/b + d	1	0.90	0.79	0.95	0.41
Positive predictive value a/a + b	1	0.74	0.38	0.88	0.32
Negative predictive value d/d + c	0.91	1	1	1	1
Positive likelihood ratio S/1 − E	0.83	10	0.76	20	0.39
Negative likelihood ratio 1 − S/E	0.17	0.11	0.79	0.05	0.43
Accuracy index (a + d)/(a + d + b + 0)	0	0.92	0.91	0.96	0.8

## Data Availability

The data presented in this study are available upon request from the corresponding author. The data are not publicly available due to ethical reasons.

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
