# Peer review of "Sensitivity and Specificity of a Screening Test for the Detection of Deficiencies in Visuo-Perceptual Skills"

_brainsci, 2024, doi:10.3390/brainsci14070705_

Round 1

Reviewer 1 Report

Comments and Suggestions for Authors

Fernando and collaborators explore whether the Petrosyan questionnaire, a screening test, could be delivered in Spanish and whether it could be useful to detect deficiencies in perceptual skills in infants. The novelty and originality of this paper are limited, but the application of the knowledge from this study could be important for both clinicians and scientists. Therefore, I am quite positive about this study.

However, there are several issues that must be considered before recommending publication:

  1. Although I am not a native English speaker, I feel that the manuscript needs proofreading.
  2. There are many typos in the manuscript, which can be quite distracting and may give the impression that the authors did not thoroughly review their work. Did the coauthors revise the final version of the manuscript before submission?
  3. Abstract: Line 23, correct the typo ("Less…").
  4. Introduction: The introduction is not linear and is sometimes hard to follow. For example, lines 34 to 40 repeat the same concept in different words. Please make an effort to improve the clarity of the introduction.
  5. Line 90: Please specify what you mean by "infants" (I assume you mean ages 8-15).
  6. Please include all important information about the Petrosyan questionnaire in your manuscript, such as its validity and reliability. For example, provide psychometric information to show that the questionnaire measures what it is intended to measure.
  7. I recommend making Figure 7 larger.
  8. Method section, line 96: You mention participants with "good visual acuity." Were they tested for this? If so, how?
  9. Line 97: Do you have the protocol number for the study, or does your ethical committee not provide one?
  10. Line 106: You mention a "valid and reliable index." Refer to point 6 and report the psychometric properties of the questionnaire.
  11. Line 110: Please cite the authors correctly (e.g., "Author L.L., Thurston T.G.").
  12. Line 111: You mention the questionnaire "applies to ages 8 to 14," but you also collected data from 15-year-olds. Please acknowledge this as a limitation of the study.
  13. Lines 131-133: Suggestion: Simplify by stating that an eye-tracker was used. (you may want to ignore this point)
  14. General comments: Often, you forget to add spaces between words.
  15. Results section: Why did you not present the correlation between the scores obtained on the questionnaire and the "gold test"? Is there a specific reason?
  16. You performed several tests. Did you correct the p-value for multiple comparisons?
  17. It is quite unusual to obtain p-values lower than 0.0001 consistently. For example, even in Table 4, where you present a 2% difference, the p-value is very small. While small differences can be significant with a large sample size, please check your analysis to ensure accuracy.
  18. Discussion section: the title is misspelled (Discusiòn)
  19. Line 186: Use the "<" notation or the exact p-value; do not mix "=" and "<" in the same context.
  20. Line 201: Check for typos.
  21. Do not include names in citations (see line 208).
  22. Consistently use either proportions or percentages (e.g., 95% or 0.92).
  23. Line 219: You state a "high value," but your specificity is 92%, not 95%. Therefore, you cannot describe it as "high" based on your definition in line 213 (a high sensitivity is greater than 95%).

I encourage the authors to revise this manuscript, as it has merit. However, in its current form, it is not suitable for publication.

Comments on the Quality of English Language
  1. Although I am not a native English speaker, I feel that the manuscript needs proofreading.

Author Response

Although I am not a native English speaker, I feel that the manuscript needs proofreading: The article was checked and corrected.

There are many typos in the manuscript, which can be quite distracting and may give the impression that the authors did not thoroughly review their work. Did the coauthors revise the final version of the manuscript before submission?: Yes, the coauthors revised the final version 

Abstract: Line 23, correct the typo ("Less…"). The full abstract has been revised and the error has been corrected.

Introduction: The introduction is not linear and is sometimes hard to follow. For example, lines 34 to 40 repeat the same concept in different words. Please make an effort to improve the clarity of the introduction. Perception is an active process involving the localization and extraction of information from the external environment. Perceptual systems organize this information, engaging in data search and retrieval to develop awareness of surroundings and self. Through perception, individuals attribute meaning to sensory input received from external senses (vision, smell, hearing, touch, and taste) and internal senses (vestibular and kinesthetic systems). The brain interprets and classifies this information, leading to the formation of both simple and complex cognitive concepts. Visual perception, notably, significantly influences learning and academic performance by aiding in the encoding, decoding, and spatial-temporal structuring of letters and information

Line 90: Please specify what you mean by "infants" (I assume you mean ages 8-15).: 200 subjects, ages 8 to 15 years

Please include all important information about the Petrosyan questionnaire in your manuscript, such as its validity and reliability. For example, provide psychometric information to show that the questionnaire measures what it is intended to measure.:  The translation process of the questionnaire Petrosyan followed the procedure established by Brislin (1970). Initially, the text was translated from the original language (American English) into Spanish by experts. Subsequently, ten Spanish-speaking optometrists, skilled in perceptual and binocular vision. They were provided with the original questionnaire in Spanish and asked to translate it back into English. The translations from all 10 optometrists were compared to the original questionnaire coincidences and non-coincidences were detected. To determine the level of similarity, criteria for interpretation and grammar were considered. Each aspect evaluated was assigned a score from 0 to 3, where: 0 = none: if the interpretation was completely different, 1 = low: if there were more different words than similar or identical words,2 = partial match: if there were more similar/identical words than different words, 3 = identical: if the words were the same. This scoring system allowed for a quantitative assessment of the fidelity of the translated versions compared to the original questionnaire.

Reference:  Brislin, R.W (1970) Back Translation for Cross-cultural Research. Journal of Cross-Cultural Psychology, 1(3) 185-216. https://doi.org/10.1177/135910457000100301

I recommend making Figure 7 larger.: Suggestion addressed

Method section, line 96: You mention participants with "good visual acuity." Were they tested for this? If so, how?:  Yes, they were tested with Snellen Chart. “Which was determined for each eye with the Snellen chart in far and near vision.”

Line 97: Do you have the protocol number for the study, or does your ethical committee not provide one?:  No, we don´t received any number. We only have a letter of acceptance

Line 106: You mention a "valid and reliable index." Refer to point 6 and report the psychometric properties of the questionnaire.: The standardization of the MVPT-4 test was administered to 2,160 individuals at 85 sites across the United States between the ages of 4 years and geriatric adult by 88 examiners. To assess reliability of test items, Cronbach´s coefficient alpha was computed for the overall sample and for each group in the standardization sample this value was 0.80. This value indicates that the MVPT-4 overall is internally consistent. To asseses validity 27 individuals were administered the MVPT-4 and TVPS-3, results indicate there is a significant correlation between both tests. Results suggest that the mvpt.4 adequately measures aspects of visual perceptual skills

Line 110: Please cite the authors correctly (e.g., "Author L.L., Thurston T.G.").: Suggestion addressed

Line 111: You mention the questionnaire "applies to ages 8 to 14," but you also collected data from 15-year-olds. Please acknowledge this as a limitation of the study: Although the primary mental habilities test (PMA) is designed for populations from 8 to 14 years old, in order to minimize the age limitation, the MVPT-4 test was used, which is designed for a wider age range

Lines 131-133: Suggestion: Simplify by stating that an eye-tracker was used. (you may want to ignore this point): We believe that in order to continue with the same format of explanation of all the procedures, we have decided to leave the explanation of the device

General comments: Often, you forget to add spaces between words.: The entire document has been revised to eliminate this error

Results section: Why did you not present the correlation between the scores obtained on the questionnaire and the "gold test"? Is there a specific reason?: Correlations were not performed because the objective of the study is to identify the sensitivity and specificity of the screening test with the gold standard. It is not our objective to correlate study variables

You performed several tests. Did you correct the p-value for multiple comparisons?: Not entirely clear observation, however, we assume that we asked the reviewer if, we corrected the p values for multiple comparisons. This action was not necessary, as the values obtained did not require it

It is quite unusual to obtain p-values lower than 0.0001 consistently. For example, even in Table 4, where you present a 2% difference, the p-value is very small. While small differences can be significant with a large sample size, please check your analysis to ensure accuracy: 

The authors valued their observation, reviewed the data in the tables again and initiated a new statistical analysis, given the small differences, in which we found new readings and corrected them in the paper. The authors considered working with the means of the results obtained in the evaluations, however, when performing the data distribution test ("Shapiro-Wilk test") and ("Kolmogorov-Smirnov test"), normality was not obtained in the data distribution that would allow us to make inferences based on the Student's t-test, therefore, we worked with medians and obtained the comparisons with the Wilcoxon test. It is possible that due to the use of medians and their very small differences that 2% was obtained between the two groups. However, the authors reviewed the information again and were able to find some values that were corrected and thus the p values of some comparisons were reevaluated and it was found that they are now in the order of p=0.01.  Table 4 shows in quantity by gender those males and females who were deficient in the evaluations, and we compared with the Wilcoxon test the results of their values of the evaluations with the test

Discussion section: the title is misspelled (Discusiòn): Suggestion addressed. Discussion

Line 186: Use the "<" notation or the exact p-value; do not mix "=" and "<" in the same context: Suggestion addressed

Line 201: Check for typos.: Suggestion addressed

Do not include names in citations (see line 208).: Suggestion addressed

Consistently use either proportions or percentages (e.g., 95% or 0.92).: Only the proportion was modified and left unchanged. 

Line 219: You state a "high value," but your specificity is 92%, not 95%. Therefore, you cannot describe it as "high" based on your definition in line 213 (a high sensitivity is greater than 95%).: there is a high specificity when it between 0.91 to 1 is greater than 95%, in this case it is 0.92 o 92%

Reviewer 2 Report

Comments and Suggestions for Authors

This study aims to evaluate the sensitivity and specificity of a screening test, i.e., the Petrosyan Visual Perceptual Questionnaire, in detecting perceptual abnormalities, and to identify any differences in perceptual skills between genders. Some considerations are listed below.

On page 2, line 75, when the Petrosyan Visual Perceptual Questionnaire is mentioned for the first time, please provide the citation for when the first version was published. 

On page 2, line 90, why is the sample described as 200 “infants” when in other sections it is stated as “n=200 students aged 8 to 15 years”?

On page 4, line 154, there is a typo with the comma.

On page 5, lines 171-172, "The Wilcoxon test for nonparametric samples was performed to identify differences in terms of gender, Table 3 shows that in all variables there is a statistically significant difference." Should this refer to Table 4? Also, there is a typo with the comma.

To validate the results of the Petrosyan Visual Perceptual Questionnaire, I suggest comparing the results from each diagnostic test (i.e., MVPT-4, PMA-SR, PMA-PS, Readalyzer, Wold copy sentence) to the results of each subtest with a comparable function.

Incorporating visualizations into the study will enhance the readers' comprehension of the results. For example, Plotting the Receiver Operating Characteristic (ROC) curve will illustrate the trade-off between sensitivity and specificity of the Petrosyan Visual Perceptual Questionnaire. A bar graph to compare perceptual skills between genders across various subtests will highlight any significant differences clearly. Or a box plot can show the distribution of scores for each subtest. Finally, scatter plots to compare the results of each diagnostic test with the corresponding subtest of the Petrosyan Visual Perceptual Questionnaire. This will help in identifying any similarities or discrepancies.

Author Response

On page 2, line 75, when the Petrosyan Visual Perceptual Questionnaire is mentioned for the first time, please provide the citation for when the first version was published. The first version of the questionnaire has not yet been published. It was presented at the virtual congress I heart VT

On page 2, line 90, why is the sample described as 200 “infants” when in other sections it is stated as “n=200 students aged 8 to 15 years”? It was a mistake that has been corrected

On page 4, line 154, there is a typo with the comma. It was a mistake that has been corrected

On page 5, lines 171-172, "The Wilcoxon test for nonparametric samples was performed to identify differences in terms of gender.  Table 3 shows that in all variables there is a statistically significant difference. " Should this refer to Table 4? Also, there is a typo with the comma. Yes, it refers to table 4. It was a mistake that has been corrected

To validate the results of the Petrosyan Visual Perceptual Questionnaire.  I suggest comparing the results from each diagnostic test (i.e., MVPT-4, PMA-SR, PMA-PS, Readalyzer, Wold copy sentence) to the results of each subtest with a comparable function. We agree. The sensitivity and specificity of each test was analyzed with the Petrosyan questionnaire and is shown in figure 1. As well, the figure 1 attend the suggestion about the comparing from the each diagnostic, however the complete questionaries evaluated by separated area

Incorporating visualizations into the study will enhance the readers' comprehension of the results. For example, Plotting the Receiver Operating Characteristic (ROC) curve will illustrate the trade-off between sensitivity and specificity of the Petrosyan Visual Perceptual Questionnaire.

 A bar graph to compare perceptual skills between genders across various subtests will highlight any significant differences clearly.  Or a box plot can show the distribution of scores for each subtest. Finally, scatter plots to compare the results of each diagnostic test with the corresponding subtest of the Petrosyan Visual Perceptual Questionnaire. This will help in identifying any similarities or discrepancies. We incorporate the ROC Curve, according to the suggestion from the revierer, it shows the confidence interval, the area under curve and the p value. Figure 2 

A graph comparing the results for each test between genders was incorporated. Figure 1

Round 2

Reviewer 2 Report

Comments and Suggestions for Authors

I don't have further comments.